# Dynamics of Entropy Production Rate in Two Coupled Bosonic Modes Interacting with a Thermal Reservoir

**DOI:** 10.3390/e24050696

**Published:** 2022-05-14

**Authors:** Tatiana Mihaescu, Aurelian Isar

**Affiliations:** 1Faculty of Physics, University of Bucharest, 077125 Magurele, Romania; mihaescu.tatiana@theory.nipne.ro; 2Department of Theoretical Physics, National Institute of Physics and Nuclear Engineering, 077125 Magurele, Romania

**Keywords:** entropy production, quantum correlations, open quantum systems, Gaussian states

## Abstract

The Markovian time evolution of the entropy production rate is studied as a measure of irreversibility generated in a bipartite quantum system consisting of two coupled bosonic modes immersed in a common thermal environment. The dynamics of the system is described in the framework of the formalism of the theory of open quantum systems based on completely positive quantum dynamical semigroups, for initial two-mode squeezed thermal states, squeezed vacuum states, thermal states and coherent states. We show that the rate of the entropy production of the initial state and nonequilibrium stationary state, and the time evolution of the rate of entropy production, strongly depend on the parameters of the initial Gaussian state (squeezing parameter and average thermal photon numbers), frequencies of modes, parameters characterising the thermal environment (temperature and dissipation coefficient), and the strength of coupling between the two modes. We also provide a comparison of the behaviour of entropy production rate and Rényi-2 mutual information present in the considered system.

## 1. Introduction

Entropy production (EP) is a basic concept in nonequilibrium classical and quantum thermodynamics. It is intimately related to the second law of thermodynamics, which enables identifying and quantifying the irreversibility of physical processes, expressed by the generation of entropy and the dissipation of heat into the surrounding environment of the systems [1,2,3,4,5,6,7,8,9,10]. According to the second law of thermodynamics, entropy change ΔS of the state of a system that exchanges energy during its interaction with a thermal environment at temperature *T* has a lower bound:(1)$$\Delta S\geq \int \frac{\delta Q}{T},$$
where δQ is the infinitesimal heat absorbed by the system. The strict inequality characterises an irreversible process for which energy is dissipated into the environment in the form of heat [11]. However, besides the entropy that flows from the system into the reservoir, some additional entropy may be intrinsically generated by the process within the system, called entropy production. From the second law of thermodynamics, it follows that EP is always non-negative; it only has a zero value when the system is in thermal equilibrium with its reservoir and it can consequently be used as a measure of the degree of irreversibility of physical processes and to characterise a broad range of nonequilibrium phenomena. Entropy production Σ can be defined as
(2)$$\Sigma \equiv \Delta S-\int \frac{\delta Q}{T}\geq 0.$$

From Equation (Equation 2), the following equality [6,12] can be derived:(3)$$\frac{dS}{dt}=\Pi \left(t\right)-\Phi \left(t\right),$$
where Π(t) denotes the irreversible EP rate, and Φ(t) the entropy flux from the system into the environment. When the system reaches a stationary state, these two quantities take strictly positive and equal values, while thermal equilibrium is reached only when both are zero. The entropy of an open system does not satisfy a continuity equation, and this prevents EP from being a physical observable; therefore, it is not generally accessible by direct probing.

The last few years showed a growing interest in studying the properties of entropies of quantum states, particularly tomographic entropies [13,14] and thermodynamic implications of quantum features, including understanding the role, properties, and evolution of EP in stochastic thermodynamics and related to the quantum theory of open systems [8,9,10,15,16,17,18].

In the case of a Markovian dynamics of open systems, described by a quantum dynamical semigroup, information monotonically flows from the system into the environment, and the corresponding EP is a non-negative quantity. Some models show a backflow of information going from the environment to the system, and this is usually interpreted as a signature of non-Markovianity. In this case, it is possible for intervals of time to exist in which EP takes negative values. However, this does not mean that the second law of thermodynamics is violated [19], but it can be understood in terms of information backflow generated by quantum non-Markovianity, which means that the system recovers a part of the information that it previously lost during interaction with the environment.

In [17], the authors carried out a study of the irreversibility generated in the stationary state of a quantum system composed of two coupled quantum oscillators, with each interacting with its local reservoir. The authors derived the expression of the rate of irreversible EP in the stationary state, and their analysis showed that the generation of correlations and EP are complementary aspects during interaction with the environment. In [18], the authors investigated the behaviour of EP rate by studying non-Markovian Brownian motion in an uncoupled bipartite quantum system interacting with two independent reservoirs. In addition, the authors in [17,18] established a connection between EP rate and quantum correlations in the bipartite system.

Here, we employ the formalism of the theory of open systems based on completely positive quantum dynamical semigroups [20] to describe the dynamics of the rate of irreversible EP in a system composed of two coupled nonresonant bosonic modes embedded in a common thermal environment by extending the study in [17,18] to some degree to analyse the time evolution of EP in this system. The influence of the environment is discussed in terms of covariance matrix by taking squeezed thermal states as initial states. We show that the evolution of EP rate strongly depends on the parameters of the initial state of the system (squeezing parameter and average thermal photon numbers of the bosonic modes), frequencies of modes, parameters characterising the thermal reservoir (temperature and dissipation coefficients), and the strength of the coupling between the two modes. Using the general expression for the rate of EP, we describe its behaviour for the initial state of the system, its time evolution, and for the non-equilibrium stationary state of the considered system. Moreover, since the correlations existing in a bipartite system are determined by its entropy, and the dynamics of correlations is thus related to EP [21,22,23,24,25,26], we provide a description of these two fundamental quantum characteristics by comparing the behaviour of EP rate and of a well-known correlation measure, namely, Rényi-2 mutual information, relative to their evolution with time and in stationary state.

The paper is organised as follows. In Section 2 we present the master Markovian equation for the density operator of an open system in interaction with a general environment, and solve the Lyapunov evolution equation for the covariance matrix of the state of the bimodal bosonic system. In Section 3 we write the expression of EP rate for Gaussian states. In Section 4 we describe the dynamics of EP rate for the considered system; in Section 5, we compare the behaviour of EP rate and Rényi-2 mutual information in the considered system. Lastly, we summarise the obtained results and present the conclusions in Section 6.

## 2. Master Equation for Two Bosonic Modes Interacting with the Environment

We study the dynamics of a system composed of two coupled bosonic modes (harmonic oscillators) in weak interaction with a thermal environment, by using the formalism based on completely positive quantum dynamical semigroups. In this framework, the Markovian irreversible time evolution of an open system is described by the Gorini–Kossakowski–Sudarshan–Lindblad master equation for density operator ρ(t) [20,27,28,29]:(4)$$\begin{array}{c} \frac{d\rho (t)}{dt}=-\frac{i}{\hbar }\left[H,\rho \left(t\right)\right]+\frac{1}{2\hbar }\sum_{j}\left(2B_{j}\rho \left(t\right)B_{j}^{†}-{\left\{\rho \left(t\right),B_{j}^{†}B_{j}\right\}}_{+}\right). \end{array}$$

Here *H* is the Hamiltonian of the open system, and operators $B_{j},B_{j}^{†},$ defined on the Hilbert space of H, describe the interaction of the system with a general environment.

The Hamiltonian of two nonresonant linearly coupled in coordinates bosonic modes of frequencies $\omega_{1}$ and $\omega_{2}$ is given by
(5)$$\begin{array}{c} H=\frac{\hbar \omega_{1}}{2}\left(x^{2}+p_{x}^{2}\right)+\frac{\hbar \omega_{2}}{2}\left(y^{2}+P_{y}^{2}\right)+qxy, \end{array}$$
where x,y,$p_{x},p_{y}$ are the dimensionless position and momentum operators of the two modes, respectively, and *q* is the coupling parameter. Operators $B_{j}$ are taken to be polynomials of the first degree in these canonical operators; if we choose initial Gaussian states, Gaussianity is preserved with time due to the linear character of the dynamics [30,31]. $R={\left\{x,p_{x},y,p_{y}\right\}}^{T}$, vector of canonically conjugated quadrature operators for the two bosonic modes; σ, 4×4 bimodal covariance matrix with elements given by the second statistical moments of the quadrature operators:(6)$$\sigma_{ij}=\mathrm{Tr}\left[\left(R_{i}R_{j}+R_{j}R_{i}\right)\rho \right],i,j=1,\ldots ,4,$$
which fully characterise the Gaussian state of a two-mode system. We neglected the first moments, since they could be transformed into zero by suitable local displacements in phase space.

The time evolution of covariance matrix σ(t) is determined by the following Lyapunov equation [29]: (7)$$\begin{array}{c} \frac{d\sigma (t)}{dt}=A\sigma \left(t\right)+\sigma \left(t\right)A^{T}+D, \end{array}$$(8)$$\begin{array}{c} A=\left(\begin{array}{cccc} -\lambda & \omega_{1} & 0 & 0\\ -\omega_{1} & -\lambda & -q & 0\\ 0 & 0 & -\lambda & \omega_{2}\\ -q & 0 & -\omega_{2} & -\lambda \end{array}\right), \end{array}$$
where *A* denotes the drift matrix, *D* is the diffusion matrix, and λ is the dissipation rate. We assume that the diffusion matrix has the following form [20,29] (we set ℏ=1):(9)$$D=2\;\mathrm{diag}\{\lambda \coth\frac{\omega_{1}}{2k_{B}T},\lambda \coth\frac{\omega_{1}}{2k_{B}T},\lambda \coth\frac{\omega_{2}}{2k_{B}T},\lambda \coth\frac{\omega_{2}}{2k_{B}T}\},$$
where $k_{B}$ is the Boltzmann constant, and *T* is the temperature of the thermal environment.

The time-dependent solution of Equation (Equation 7) is [29]
(10)$$\begin{array}{c} \sigma \left(t\right)=M\left(t\right)\left[\sigma \left(0\right)-\sigma_{s}\right]M^{T}\left(t\right)+\sigma_{s}, \end{array}$$
where M(t)≡exp(At).

This evolution, generated by a Gaussian completely positive map, is determined by the 4 × 4 real matrices *M* and $Y=\sigma_{s}-M\sigma_{s}M^{T}$, which satisfy $Y+i\Omega \geq iM\Omega M^{T}$, where Ω is the symplectic matrix
(11)$$\Omega =\overset{2}{\underset{1}{⨁}}\left(\begin{array}{cc} 0 & 1\\ -1 & 0 \end{array}\right).$$

Since unitary evolution generated by Hamiltonian (Equation 5) does not commute with dynamics generated by the interaction of the system with the thermal bath, the coupling between the two bosonic modes affects irreversibility, while the open system evolves (in our case, in the limit of large times) into a nonequilibrium steady state, described by stationary covariance matrix $\sigma_{s},$ which can be obtained by setting $\frac{d\sigma (t)}{dt}=0$ in Equation (Equation 7):(12)$$A\sigma_{s}+\sigma_{s}A^{T}=-D.$$

Pair of matrices *A* and *D* represent the drift matrix and diffusion matrix of a quantum system if and only if $D+iA\Omega^{T}-i\Omega A^{T}\geq 0$, derived from the uncertainty principle [32]. In order for the system to be stable, i.e., to admit a steady state, condition $A+A^{T}<0$ needs to be satisfied.

The model is thus analytically solvable, and covariance matrix σ(t) depends on the initial state, on parameters of the system and thermal bath, and on coupling between modes. The chosen form for the diffusion matrix leads, in the case of noncoupled bosonic modes q=0, to an asymptotic Gibbs state that describes a thermal equilibrium with the environment [20,25]. If the modes are coupled, the stationary state is not a product state anymore. We provide here its form in the simple resonant modes case ($\omega_{1}=\omega_{2}=\omega$) (we set $k_{B}=1$):(13)$$\begin{array}{c} \sigma_{s}\left(\infty \right)=\frac{\coth\left(\frac{\omega }{2T}\right)}{2\left(L^{2}-q^{2}\omega^{2}\right)}\\ \times \left(\begin{array}{cccc} 2L^{2}-q^{2}\omega^{2} & \lambda q^{2}\omega & -q\omega L & -\lambda qL\\ \lambda q^{2}\omega & 2L^{2}+q^{2}\left(\lambda^{2}-2\omega^{2}\right) & -\lambda qL & q\omega (L-q^{2})\\ -q\omega L & -\lambda qL & 2L^{2}-q^{2}\omega^{2} & \lambda q^{2}\omega \\ -\lambda qL & q\omega (L-q^{2}) & \lambda q^{2}\omega & 2L^{2}+q^{2}\left(\lambda^{2}-2\omega^{2}\right) \end{array}\right), \end{array}$$
where $L\equiv \omega^{2}+\lambda^{2}$. When q=0, then Equation (Equation 13) becomes $\sigma_{G}\left(\infty \right)=\coth\left(\frac{\omega }{2T}\right)I.$

## 3. Entropy Production Rate for Gaussian States

The usual approach for studying the EP is based on von Neumann entropy. The dynamics of the open quantum systems given by the master Equation (Equation 4) can be reformulated in terms of the Fokker-Plank equation for Wigner distribution function, therefore it is appropriate to describe the evolution of EP by using a corresponding approach based on the phase space formalism [6,33,34]. Consequently, we introduce the Wigner EP rate [12], given by
(14)$$\Pi \left(t\right)\equiv -\partial_{t}K(W\left(t\right)||W_{s}),$$
where $K(W\left(t\right)||W_{s})$ is Wigner relative entropy, W(t) is the time-dependent Wigner function and $W_{s}$ is Wigner function for the stationary state.

We introduce the symplectic matrix representing time reversal operator E=diag(1,−1,1,−1). Then, dynamic variables can be divided according to their time symmetry. Drift matrix *A* (Equation 8) is split into an irreversible component $A^{\mathrm{irr}}$, given by $A^{\mathrm{irr}}=\frac{1}{2}\left(A+EAE^{T}\right)$, and a reversible one $A^{\mathrm{rev}}=\frac{1}{2}\left(A-EAE^{T}\right)$:(15)$$A^{\mathrm{irr}}=\mathrm{diag}\left(-\lambda ,-\lambda ,-\lambda ,-\lambda \right),$$
(16)$$A^{\mathrm{rev}}=\left(\begin{array}{cccc} 0 & \omega_{1} & 0 & 0\\ -\omega_{1} & 0 & -q & 0\\ 0 & 0 & 0 & \omega_{2}\\ -q & 0 & -\omega_{2} & 0 \end{array}\right).$$

The analytical expression of EP rate Π(t) as a function of drift matrix *A*, diffusion matrix *D*, and covariance matrix σ is the following [6,17,18]:(17)$$\Pi \left(t\right)=\frac{1}{2}\mathrm{Tr}\left[\sigma^{-1}\left(t\right)D\right]+2\mathrm{Tr}\left[A^{\mathrm{irr}}\right]+2\mathrm{Tr}\left[{\left(A^{\mathrm{irr}}\right)}^{T}D^{-1}A^{\mathrm{irr}}\sigma (t)\right].$$

In particular, when the system reaches nonequilibrium stationary state $\sigma_{s}$, expression (Equation 17) becomes [17]
(18)$$\Pi_{s}=\mathrm{Tr}\left[A^{\mathrm{irr}}\right]+2\mathrm{Tr}\left[{\left(A^{\mathrm{irr}}\right)}^{T}D^{-1}A^{\mathrm{irr}}\sigma_{s}\right].$$

## 4. Dynamics of Entropy Production Rate

The degree of irreversibility of the dynamics of an open system is associated with the EP rate of its state. We now describe the dynamics of the EP rate in terms of the coupling between two bosonic modes and parameters characterising the initial Gaussian state of the considered system and thermal reservoir.

### 4.1. Initial Entropy Production Rate

We consider an initial squeezed thermal state with covariance matrix
(19)$$\sigma_{0}=\left(\begin{array}{cccc} a & 0 & c & 0\\ 0 & a & 0 & -c\\ c & 0 & b & 0\\ 0 & -c & 0 & b \end{array}\right),$$
where
(20)$$\begin{array}{c} a=2n_{1}\cosh^{2}r+2n_{2}\sinh^{2}r+\cosh2r,\\ b=2n_{1}\sinh^{2}r+2n_{2}\cosh^{2}r+\cosh2r,\\ c=\left(n_{1}+n_{2}+1\right)\sinh2r. \end{array}$$

$n_{1}$ and $n_{2}$ are the average thermal photon numbers of the modes, and *r* is the squeezing of the initial state.

Explicit calculations performed in Equation (Equation 17) lead to the following expression of EP rate at the initial moment of time:(21)$$\begin{array}{c} \Pi \left(0\right)=2\lambda [-4+\frac{1}{\left(1+2n_{1}\right)\left(1+2n_{2}\right)}\\ \times (\left(-n_{1}+n_{2}+\left(1+n_{1}+n_{2}\right)\cosh2r\right)\coth\frac{\omega_{1}}{2T}\\ +\left(n_{1}-n_{2}+\left(1+n_{1}+n_{2}\right)\cosh2r\right)\coth\frac{\omega_{2}}{2T})\\ \times \left(\left(1+2n_{1}\right)\left(1+2n_{2}\right)+\coth\frac{\omega_{1}}{2T}\coth\frac{\omega_{2}}{2T}\right)\tanh\frac{\omega_{1}}{2T}\tanh\frac{\omega_{2}}{2T}]. \end{array}$$

For an initial squeezed vacuum state ($n_{1}=n_{2}=0$), expression (Equation 21) becomes
(22)$$\begin{array}{c} \Pi_{v}\left(0\right)=2\lambda \left[-4+\cosh2r\left(\coth\frac{\omega_{1}}{2T}+\coth\frac{\omega_{2}}{2T}+\tanh\frac{\omega_{1}}{2T}+\tanh\frac{\omega_{2}}{2T}\right)\right], \end{array}$$
which simplifies in the resonant case ($\omega_{1}=\omega_{2}\equiv \omega$):(23)$$\begin{array}{c} \Pi_{\mathrm{vr}}\left(0\right)=8\lambda \left[\cosh2r\coth\frac{\omega }{T}-1\right]. \end{array}$$

For resonant modes and an initial symmetric thermal state (r=0), EP rate has the form
(24)$$\begin{array}{c} \Pi_{T}\left(0\right)=\frac{8\lambda {\left(1+n-ne^{\frac{\omega }{T}}\right)}^{2}\left[\coth\frac{\omega }{T}-1\right]}{1+2n}. \end{array}$$

The simplest expression is obtained for an initial coherent state:(25)$$\begin{array}{c} \Pi_{c}\left(0\right)=8\lambda \left[\coth\frac{\omega }{T}-1\right]. \end{array}$$

EP rate at the initial moment of time does not depend on coupling *q* between modes. The initial EP rate increases with squeezing *r* and dissipation rate λ. In addition, for an initial symmetric squeezed vacuum state and coherent state, the rate of EP also increases with the temperature of the reservoir. Equation (Equation 25) shows that the minimal zero value is reached in the case of a coherent state for zero reservoir temperature.

Figure 1 and Figure 2 illustrate the dependence of EP rate at the initial moment of time on parameters characterising the initial Gaussian state and the thermal reservoir. Figure 1a shows that, for an initial squeezed vacuum state, EP rate $\Pi_{v}\left(0\right)$ decreases by increasing both frequencies of the two bosonic modes. In addition, the state of nonresonant modes generally manifests a larger rate of EP, compared to the resonant case. This behaviour could be interpreted as a result of the breaking of the symmetry between the two subsystems, which is a factor leading to the increase in EP rate. Figure 1b illustrates for the particular case of an initial symmetric squeezed thermal state that the initial EP rate increases with both squeezing of modes and dissipation. Squeezing, which is a useful resource in both quantum information and quantum thermodynamics, is intimately related to the Heisenberg uncertainty principle; by introducing an asymmetry between position and momentum uncertainties, it modifies energy fluctuations, introduces extra increase in entropy, so EP is also modified [35,36,37,38,39]. Figure 2a,b show that, for an initial symmetric squeezed thermal state in resonant case, Π(0) slightly increases with both the average thermal photon number and the frequency of modes for relatively small temperatures of the thermal bath, while it decreases by increasing the photon number and frequency of modes for larger temperatures. Π(0) decreases by increasing the temperature of reservoir for relatively small values of the temperature; for larger values, it increases with temperature. This behaviour is the result of the competition between influences exerted by parameters characterising the initial state (thermal photon number and frequency of modes) and the environment (temperature and dissipation) on EP rate.

### 4.2. Time Evolution of Entropy Production Rate

We now analyse the behaviour of EP rate as a function of time, coupling between two bosonic modes, and parameters characterising the initial state of the system and thermal reservoir. Its general analytical expression is too intricate to be provided here; we thus only report the expression of EP rate in the case of uncoupled bosonic modes (q=0) for an initial squeezed vacuum state in the resonant case ($\omega_{1}=\omega_{2}\equiv \omega$):(26)$$\begin{array}{c} \Pi \left(t\right)=4\lambda [-1+e^{-2t\lambda }\left(-1+\cosh2r\tanh\frac{\omega }{2T}\right)\\ +\left(e^{2t\lambda }\left(\left(-1+e^{2t\lambda }\right)\left(1+\cosh\frac{\omega }{T}\right)+\cosh2r\sinh\frac{\omega }{T}\right)\right)/\\ \left(e^{2t\lambda }\left(-2+e^{2t\lambda }\right)+\left(2-2e^{2t\lambda }+e^{4t\lambda }\right)\cosh\frac{\omega }{T}+2\left(-1+e^{2t\lambda }\right)\cosh2r\sinh\frac{\omega }{T}\right)]. \end{array}$$

The time evolution of EP rate Π(t) is illustrated in Figure 3 for an initial symmetric squeezed thermal state (Equation 19), (Equation 20) as a function of the squeezing of initial state, in cases of (a) uncoupled and (b) coupled nonresonant bosonic modes. We can see that Π(t) is always positive in the considered Markovian approximation and it decreases with time. At a given moment, Π(t) increases with the squeezing of initial state. Squeezing introduces an asymmetry between position and momentum uncertainties of modes, which modifies energy fluctuations and introduces an extra increase in entropy, thereby leading to an EP change [35,36,37,38,39].

The time evolution of the EP rate Π(t) is illustrated in Figure 4 for an initial symmetric thermal state as a function of the temperature of reservoir, in cases of (a) uncoupled and (b) coupled nonresonant bosonic modes. Π(t) decreases by increasing the temperature of a thermal reservoir for relatively small values of temperature; for larger values, it increases with temperature. This behaviour at a given moment of time is the result of the competition between influences provided by the thermal photon number of modes and the temperature of thermal bath on the EP rate.

Figure 5 illustrates the time evolution of EP rate Π(t) as a function of the dissipation parameter, for an initial coherent state, in the case of (a) resonant and (b) nonresonant modes. Π(t) increases with the dissipation rate of the environment, and this EP rate behaviour is a signature of the increase in the degree of irreversibility with losses generated during the interaction of the considered system with its reservoir.

The time evolution of EP rate Π(t) as a function of the coupling between the two bosonic modes, in cases of resonant and nonresonant modes, is shown in Figure 6a,b for an initial symmetric squeezed thermal state and in Figure 7a,b for an initial coherent state. The values of working parameters were chosen in agreement with the Markovian condition of weak coupling λ of the system to the environment and with stability condition $A+A^{T}<0$. Under these assumptions, Π(t) is increasing with coupling *q* between modes. Therefore, the stronger the coupling between modes is, the more irreversible the corresponding stationary process is. Coupling between modes is crucial relative to EP rate in the stationary state. Indeed, for zero coupling between modes, EP rate is zero in stationary state, when the system is actually in thermal equilibrium with the reservoir. For nonzero coupling between modes, on the other hand, Π(t) asymptotically tends with time to a nonzero value in the nonequilibrium stationary state. From the above, it follows that, depending on the considered parameters, the time evolution of EP is monotonous and may also present oscillations that are relatively denser and more intense in the case on nonresonant modes compared to the resonant case.

### 4.3. Entropy Production Rate in Stationary State

The general expression of the EP rate in the stationary state is
(27)$$\begin{array}{c} \Pi_{s}=[\lambda q^{2}(q^{2}\left(\omega_{1}^{4}+18\omega_{1}^{2}\omega_{2}^{2}+\omega_{2}^{4}+10\lambda^{2}\left(\omega_{1}^{2}+\omega_{2}^{2}\right)\right)-16\omega_{1}\omega_{2}\left(\lambda^{2}+\omega_{1}^{2}\right)\left(\lambda^{2}+\omega_{2}^{2}\right)\\ +(16\lambda^{6}+20\lambda^{4}\left(\omega_{1}^{2}+\omega_{2}^{2}\right)+\omega_{1}\omega_{2}\left(-3q^{2}+4\omega_{1}\omega_{2}\right)\left(\omega_{1}^{2}+\omega_{2}^{2}\right)\\ +\lambda^{2}\left(4\omega_{1}^{4}-6q^{2}\omega_{1}\omega_{2}+24\omega_{1}^{2}\omega_{2}^{2}+4\omega_{2}^{4}\right))\\ \times \left({\left(\coth\frac{\omega_{1}}{2T}\right)}^{2}+{\left(\coth\frac{\omega_{2}}{2T}\right)}^{2}\right)\tanh\frac{\omega_{1}}{2T}\tanh\frac{\omega_{1}}{2T})]/\\ \lambda^{2}+\omega_{1}^{2}{)}^{2}+4q^{2}\omega_{1}\omega_{2}+2\left(4\lambda^{2}-\omega_{1}^{2}\right)\omega_{2}^{2}+\omega_{2}^{4})\\ \times (\lambda^{4}+\omega_{1}\omega_{2}\left(\omega_{1}\omega_{2}-q^{2}\right)+\lambda^{2}\left(\omega_{1}^{2}+\omega_{2}^{2}\right))]. \end{array}$$

In the resonant case, it simplifies to the following form:(28)$$\Pi_{sr}=\frac{q^{2}\lambda \left(\lambda^{2}+\omega^{2}\right)}{{\left(\lambda^{2}+\omega^{2}\right)}^{2}-q^{2}\omega^{2}}.$$

EP rate in the stationary state does not depend on the initial state, as expected. In addition, in the resonant case, $\Pi_{sr}$ also does not depend on reservoir temperature. In particular, for ω=1, an expression similar to Equation (Equation 12) in [17] is obtained. The open system composed of two resonant bosonic modes has symmetric configuration; the corresponding nonequilibrium stationary state to which it evolves has the most minimal value of the EP rate, and it is the closest possible to an equilibrium state [17].

Figure 8a,b and Figure 9a,b illustrate the dependence of $\Pi_{s}$ on frequency and coupling between bosonic modes and parameters characterising the thermal reservoir, i.e. dissipation and temperature. EP rate in the stationary state increases with the coupling between modes and with dissipation, while it decreases by increasing the frequency of the modes in the resonant case. In the nonresonant case, it slightly increases with the temperature of the thermal bath for relatively small values, and it saturates for larger values of temperature. If the two modes are uncoupled (q=0), from Equation (Equation 32) it follows that the EP rate in the stationary state vanishes, and the system relaxes from the nonequilibrium stationary state toward equilibrium Gibbs thermal state, in agreement with previously obtained results [15,18,40,41].

## 5. Entropy Production and Dynamics of Gaussian Rényi-2 Mutual Information

Gaussian state ρ of a two-mode continuous variable system can be described by positive Wigner distribution in phase space [42]
(29)$$W_{\rho }\left(\xi \right)=\frac{1}{\pi^{2}\sqrt{\det\sigma }}\exp\left(-\xi^{⊺}\sigma^{-1}\xi \right),$$
where $\xi \in R^{4}$ and σ is the covariance matrix completely characterising the Gaussian state.

Entropy is usually quantified by using von Neumann entropy. An alternative quantifier of the quantum information contained in a Gaussian state is Shannon entropy of Wigner distribution (Equation 29):(30)$$S_{\sigma }\left(\rho \right)=\frac{1}{2}\ln\left(\det\;\sigma \right)+2\left(1+\ln\pi \right).$$

In addition, Rényi-α entropies were introduced in quantum information theory that form a family of additive entropies related to derivatives of the free energy with respect to temperature, defined by:(31)$$S_{\alpha }\left(\rho \right)={\left(1-\alpha \right)}^{-1}\ln\left(\mathrm{Tr}\rho^{\alpha }\right),\;\alpha \geq 0.$$

Rényi entropies represent useful instruments for studying quantum correlations in multipartite systems. Up to an additive constant, expression (Equation 30) coincides with the Rényi entropy of order 2, given by Equation (Equation 31) for α=2. For α=1 Rényi entropy becomes von Neumann entropy $S_{1}\left(\rho \right)=-\mathrm{Tr}\left(\rho \ln\rho \right)$, and for α=2 from expression (Equation 31) we obtain $S_{2}\left(\rho \right)=-\ln\left(\mathrm{Tr}\;\rho^{2}\right)$, which is the opposite of the logarithm of purity of the state ρ. Using Equation (Equation 29), we obtain the following expression of the Rényi-2 entropy for Gaussian states [43]:(32)$$S_{2}\left(\rho \right)=\frac{1}{2}\ln\left(\det\sigma \right).$$

For pure states (detσ=1) $S_{2}\left(\rho \right)=0$ and it increases with the mixedness of the state. By comparing the expressions in Equations (Equation 30) and (Equation 32), we see that Rényi and Shannon entropy indeed coincide, up to an additional constant. Rényi-2 entropy has all the required properties to be a legitimate measure of entropy, including strong subadditivity [43].

For any bipartite Gaussian state ρ of a system with subsystems *A* and *B*, Gaussian Rényi-2 mutual information is defined by
(33)$$I\left(\rho_{A:B}\right)=S_{2}\left(\rho_{A}\right)+S_{2}\left(\rho_{B}\right)-S_{2}\left(\rho \right),$$
where $\rho_{A}$ and $\rho_{B}$ are the two marginals of ρ. Two-mode covariance matrix σ is written in block form that contains covariance matrices of the parties:(34)$$\sigma =\left(\begin{array}{cc} \sigma_{A} & \sigma_{C}\\ \sigma_{C}^{T} & \sigma_{B} \end{array}\right),$$
the Gaussian Rényi-2 mutual information has expression [43]
(35)$$I\left(\rho_{A:B}\right)=\frac{1}{2}\ln\left(\frac{\det\sigma_{A}\det\sigma_{B}}{\det\sigma }\right).$$

$I\left(\rho_{A:B}\right)\geq 0$ and it represents a measure of the total quadrature correlations between the parties *A* and *B* in the state ρ.

The analytical time-dependent expression of Rényi-2 mutual information for the system considered in this paper is very complicated; therefore, we only report its expression for the stationary state in the particular case of resonant modes:(36)$$I_{s}=\frac{1}{2}\ln\frac{\left(\lambda^{2}+\omega \left(\omega -q\right)\right)\left(\lambda^{2}+\omega \left(\omega +q\right)\right)\left(q^{4}+8q^{2}\left(\lambda^{2}-\omega^{2}\right)+16{\left(\lambda^{2}+\omega^{2}\right)}^{2}\right)}{4{\left(q^{2}\left(\lambda^{2}-\omega^{2}\right)+2{\left(\lambda^{2}+\omega^{2}\right)}^{2}\right)}^{2}}.$$

This does not depend on temperature.

We now describe the time evolution of the Gaussian Rényi-2 mutual information and its behaviour in the stationary state, and compare it with EP rate behaviour.

The time evolution of mutual information I(t) is illustrated in Figure 3c as a function of the squeezing of the initial state, which decreases with time. For a given moment of time, I(t) increases with the squeezing of initial state, such as EP rate Π(t).

The time evolution of mutual information I(t) is illustrated in Figure 4c as a function of the reservoir temperature. It decreases by increasing the thermal bath temperature. For relatively large temperature values, EP rate Π(t) manifested similar behaviour.

Figure 5c illustrates the time evolution of mutual information I(t) as a function of the dissipation parameter for an initial coherent state. In comparison with EP rate Π(t), it generally decreases by increasing the dissipation rate of the environment. In addition, I(t) is zero at the initial moment of time; after that, it has nonzero values, and its generation is due to the coupling between modes.

The time evolution of mutual information I(t) as a function of the coupling between two bosonic modes is illustrated in Figure 6c and Figure 7c. It increased with coupling *q* between modes, as expected. Similar behaviour relative to Π(t) was observed. Therefore, the stronger correlations between modes are, the more irreversible the corresponding evolution and stationary process are. The coupling between modes is crucial to EP rate $\Pi_{s}$ in the stationary state; for zero coupling between modes $\Pi_{s}=0$ in the stationary state, when the system is actually in thermal equilibrium with the environment. The same result is valid for mutual information, which tends asymptotically to zero for large times. In contrast, for nonzero coupling between modes, EP rate and mutual information asymptotically tend with time to a nonzero value in nonequilibrium stationary state.

Figure 8c,d show that, in the nonresonant case, mutual information in the stationary state, like the EP rate, slightly increases with temperature for relatively small values, and it saturates for larger temperature values.

Like the EP rate, mutual information in the stationary state increases with the coupling between modes, as illustrated in Figure 8c. More correlations between the two bosonic modes are shared, larger the irreversibility is, that is larger the entropy generated in the stationary state is. In particular, Figure 8a,c and Figure 9 show that, in the case of two uncoupled bosonic modes (q=0), when they separately reach locally thermal equilibrium states, EP rate and mutual information vanish.

Figure 8d shows that, different from the behaviour of EP rate illustrated in Figure 8b, mutual information decreases by increasing dissipation, as expected, due to the destructive effect of the environment.

This section’s results confirm the conclusions presented in [17,18] concerning the relationship between EP rate and correlations existing between two modes of the considered system.

## 6. Summary and Conclusions

We described the Markovian time evolution of entropy production rate as a measure of irreversibility manifested in a bipartite quantum system composed of two coupled bosonic modes immersed in a common thermal environment. The dynamics of the system was studied in the framework of the formalism of theory of open systems based on completely positive quantum dynamical semigroups for initial two-mode squeezed thermal states, squeezed vacuum states, thermal states, and coherent states. We also described time evolution and behaviour in the stationary state of the measure of total correlations shared between two modes, namely, Gaussian Rényi-2 mutual information, and compared them with the behaviour of the entropy production rate.

The behaviour and evolution of the rate of entropy production, and total correlations strongly depend on the parameters of the initial Gaussian state (squeezing parameter and average thermal photon numbers), frequencies of modes, parameters characterising the thermal environment (temperature and dissipation rate), and the strength of coupling between the two modes.

In the case of the Markovian dynamics of open quantum systems, information flows from the system into the environment and the rate of entropy production is correspondingly a positive quantity. The main results of our investigation are summarised as follows:-Entropy production rate increases with the squeezing between modes and with dissipation rate; its time evolution is monotonous and may also present oscillations that are relatively more dense and intense in the case of nonresonant modes compared to the resonant case. Squeezing introduces asymmetry between position and momentum uncertainties of modes that modifies energy fluctuations and introduces an additional increase in entropy; this leads to an increase in entropy production rate. The increase in entropy production rate with dissipation can be interpreted as a signature of the increase in degree of irreversibility with losses generated during the interaction of the system under scrutiny with the thermal reservoir. In comparison, mutual information increases with the squeezing of the initial state, like the entropy production rate, while it decreases by increasing the dissipation rate of the environment, in contrast to the entropy production rate.-Entropy production rate decreases by increasing the reservoir temperature for relatively small temperature values, while it increases with temperature for larger values; this behaviour is the result of the competition between influences produced by parameters characterising the initial state and bath temperature on the entropy production rate. Similarly, mutual information decreases by increasing temperature of thermal environment.-At the initial moment of time, entropy production rate does not depend on coupling between modes. For an initial symmetric squeezed vacuum state and a coherent state, the initial entropy production rate increases with reservoir temperature, and the minimal value of zero is reached in the case of a coherent state for zero reservoir temperature.-Entropy production rate and mutual information increase with the coupling between modes. Consequently, the stronger the coupling between the modes, and therefore the stronger their correlations, the more irreversible is the corresponding evolution and stationary process, that is the larger the entropy generated during the interaction of the system with its environment. Coupling is crucial relatively to these quantities in the stationary state: if coupling between the modes tends to zero, then the entropy production rate tends to zero in the stationary state, and the system relaxes from a non-equilibrium stationary state toward the equilibrium Gibbs thermal state. The same result is valid for mutual information, which tends asymptotically to zero for large times, for uncoupled modes. By contrast, for nonzero coupling between the modes, entropy production rate and mutual information tend asymptotically with time to a nonzero value in the non-equilibrium stationary state.-Entropy production rate in the stationary state increases with the coupling between modes, with dissipation, and slightly with the temperature of the thermal bath for relatively small values; it saturates for larger values of temperature (in the nonresonant case), while it decreases by increasing the frequency of the modes in the resonant case. Entropy production rate in the stationary state does not depend on the initial state; in addition, in the resonant case, it does not also depend on reservoir temperature. In the nonresonant case, mutual information in the stationary state, like the entropy production rate, slightly increases with temperature for relatively small values, and it saturates for larger temperature values. In the stationary state, mutual information increases with the coupling between modes, like the entropy production rate. Different from the behaviour of the entropy production rate, mutual information decreases by increasing dissipation, as expected, due to the destructive effect of the environment.

Obtained results in this paper confirm and are complementary to those in [17,18], emphasising the closed relation between irreversibility that quantifies the difference from reversible quasistatic transformations generated by the dynamical and stationary process, and correlations existing in the considered bipartite system.

In order to extend the present analysis, we plan to take into consideration the role played by the squeezing in the thermal reservoir, representing a quantum thermodynamic resource, and perform a similar investigation of the dynamics of entropy production rate in a bipartite system interacting with a squeezed thermal reservoir [10,38,44], which manifests additional thermodynamic features compared to the thermal reservoir. 

## Figures and Tables

**Figure 1 entropy-24-00696-f001:**
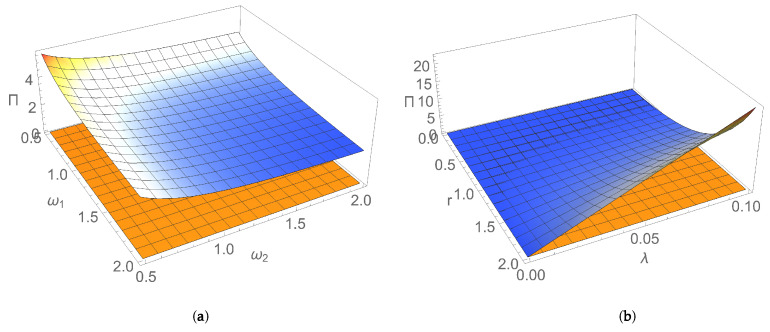
Entropy production rate versus: (**a**) frequencies $\omega_{1}$, $\omega_{2}$ of the bosonic modes for an initial squeezed vacuum state with squeezing r=1 and for dissipation λ=0.1; (**b**) squeezing *r* and dissipation λ for an initial symmetric squeezed thermal state with thermal photon number n=1 and for frequency ω=1 of the resonant modes. In both graphs, temperature is T=1.

**Figure 2 entropy-24-00696-f002:**
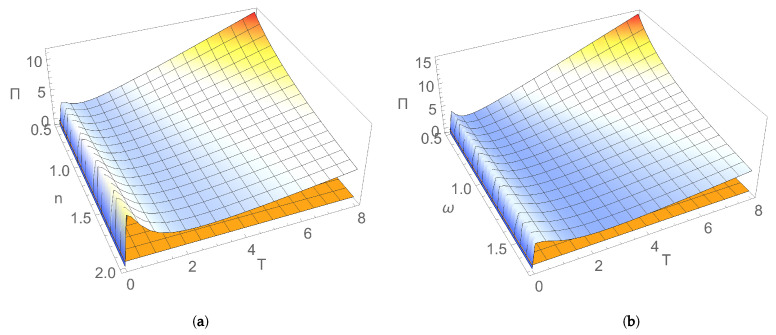
(**a**) Entropy production rate for an initial symmetric squeezed thermal state with resonant modes versus: (**a**) average thermal photon number *n* and temperature *T*, for frequency ω=1; (**b**) frequency ω and temperature *T*, for the average thermal photon number n=1. In both graphs one sets squeezing r=1 and dissipation λ=0.1.

**Figure 3 entropy-24-00696-f003:**
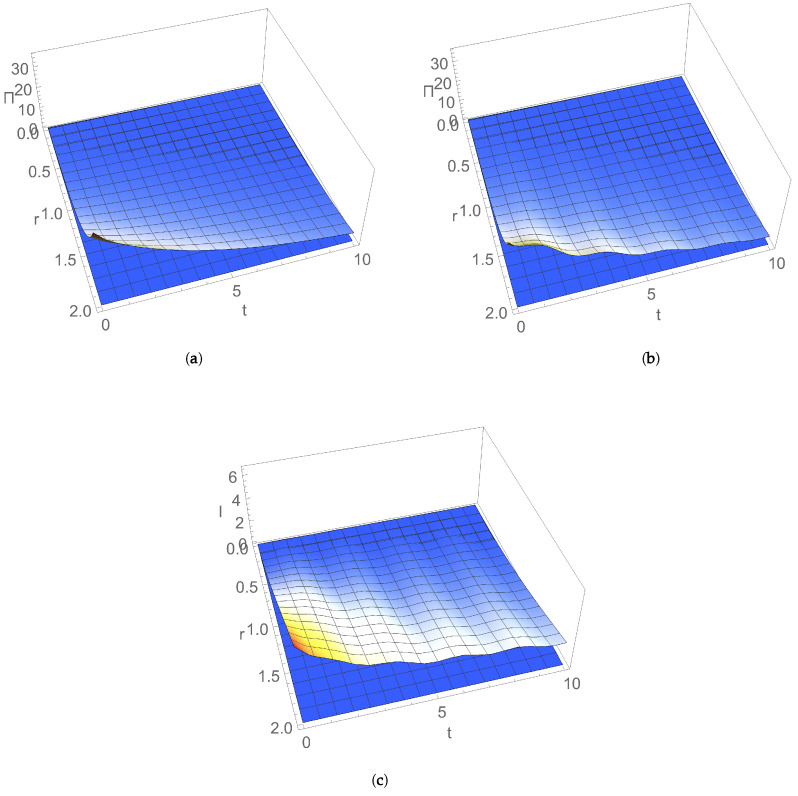
Dependence on time *t* and squeezing *r* for nonresonant modes with frequencies $\omega_{1}=1$, $\omega_{2}=1.7$ of entropy production rate for dissipation λ=0.1, (**a**) uncoupled modes with q=0 and (**b**) coupled modes with q=0.2; (**c**) mutual information for q=0.2 and dissipation λ=0.1. Initial state is a symmetric squeezed thermal state with average thermal photon number n=1 and thermal bath temperature T=0.1.

**Figure 4 entropy-24-00696-f004:**
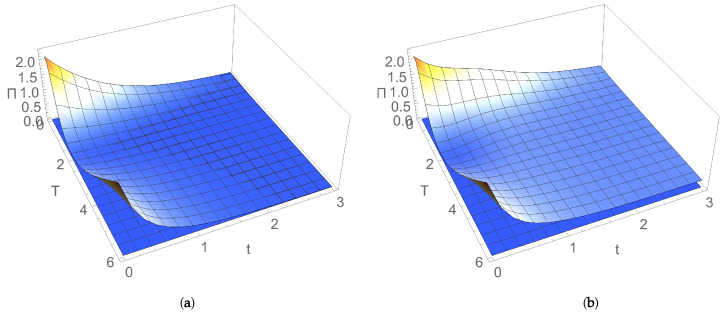
Dependence on time *t* and temperature *T* for nonresonant modes with frequencies $\omega_{1}=1$, $\omega_{2}=1.7$ of entropy production rate for (**a**) uncoupled modes with q=0 and (**b**) coupled modes with q=0.8 for an initial symmetric thermal state (squeezing r=0) and average thermal photon number n=1; (**c**) mutual information for q=0.8 for an initial symmetric squeezed thermal state with squeezing r=1 and average thermal photon number n=1. Dissipation parameter is λ=0.4.

**Figure 5 entropy-24-00696-f005:**
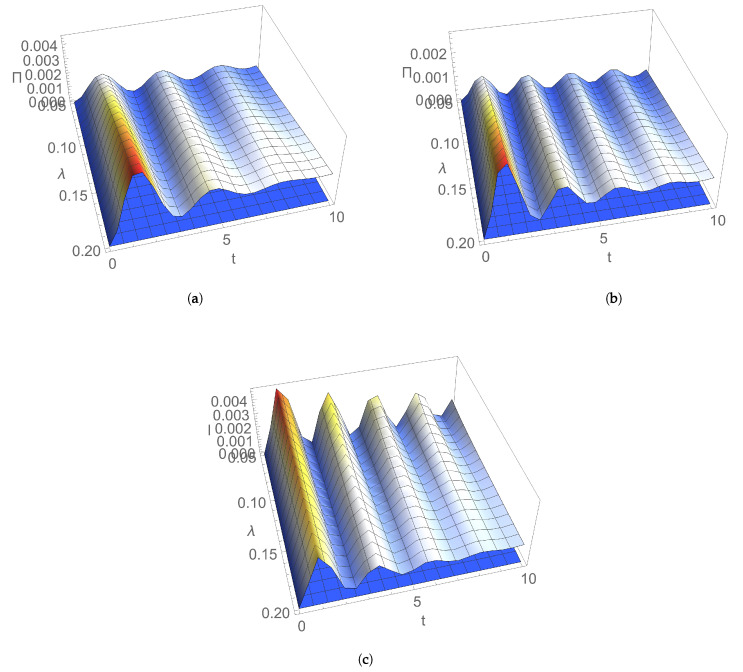
Dependence on time *t* and dissipation λ for an initial coherent state of entropy production rate for (**a**) resonant modes with frequency ω=1 and (**b**) nonresonant modes with frequencies $\omega_{1}=1,\omega_{2}=1.7$; (**c**) mutual information for nonresonant modes with frequencies $\omega_{1}=1,\omega_{2}=1.7$. Coupling constant q=0.1, and temperature T=0.1.

**Figure 6 entropy-24-00696-f006:**
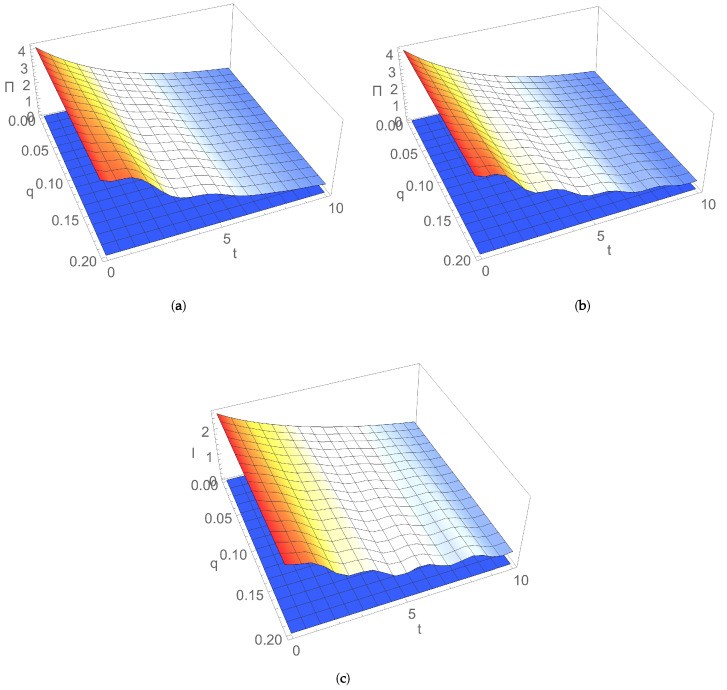
Dependence on time *t* and coupling *q* between modes of entropy production rate for (**a**) resonant modes with frequency ω=1 and (**b**) nonresonant modes with frequencies $\omega_{1}=1,\omega_{2}=1.7$; (**c**) mutual information for nonresonant modes with frequencies $\omega_{1}=1,\omega_{2}=1.7$. Initial state is a symmetric squeezed thermal state with squeezing r=1 and average thermal photon number n=1. Thermal bath parameters: temperature T=0.1 and dissipation λ=0.1.

**Figure 7 entropy-24-00696-f007:**
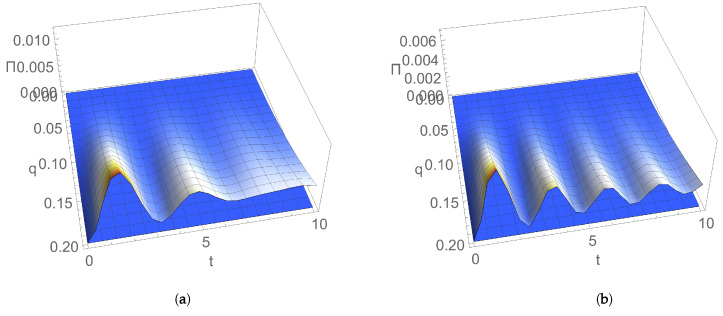
Dependence on time *t* and coupling *q* between modes for an initial coherent state of entropy production rate for (**a**) resonant modes with frequency ω=1 and (**b**) nonresonant modes with frequencies $\omega_{1}=1,\omega_{2}=1.7$; (**c**) mutual information for nonresonant modes with frequencies $\omega_{1}=1,\omega_{2}=1.7$. Thermal bath parameters: temperature T=0.1 and dissipation λ=0.1.

**Figure 8 entropy-24-00696-f008:**
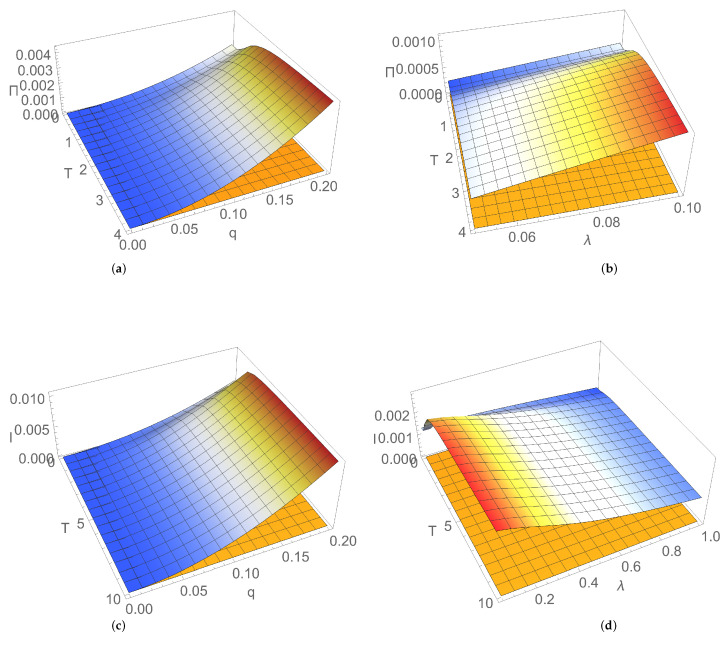
Dependence on reservoir temperature *T* and coupling *q* between modes (left) and (right) dissipation λ in stationary state of entropy production rate for (**a**) λ=0.1 and (**b**) q=0.1; mutual information, for (**c**) λ=0.1 and (**d**) q=0.1. Mode frequencies: $\omega_{1}=1,\omega_{2}=1.7$.

**Figure 9 entropy-24-00696-f009:**
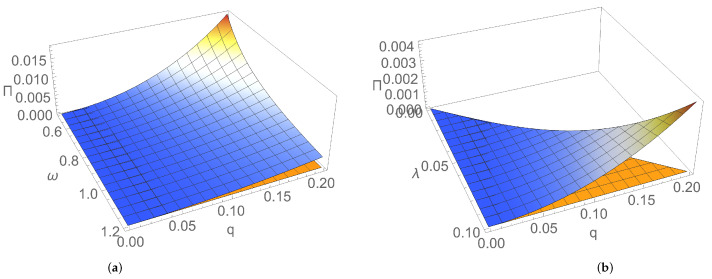
Entropy production rate in stationary state versus coupling *q* and (**a**) frequency of resonant modes ω for dissipation λ=0.1; (**b**) dissipation λ for frequency of resonant modes ω=1.

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
