# Peer review of "Dynamics of Entropy Production Rate in Two Coupled Bosonic Modes Interacting with a Thermal Reservoir"

_entropy, 2022, doi:10.3390/e24050696_

Round 1
Reviewer 1 Report
Paper is interesting, contains new results on the entropy production for bosonic modes interacting with invironments. The dynamics of the process is described by the Gorini-Kossakowski-Sudarshan-Lindblad equation. Special attention is devoted to Gaussian states of bosonic modes. This paper is clearly written and contains some novelty. I recommend to publish this paper as it is.
Author Response
We appreciate the positive feedback from the reviewer and we would like to thank him very much for his recommendation.
Reviewer 2 Report
In this manuscript, the authors analyse the dynamics of the rate of irreversible entropy production in a system consisting of two coupled non-resonant bosonic modes embedded in a common thermal environment.
The subject is somehow interesting and the used approach is sceintifically sound. However, overall the work done appears of lightweight.
It could be strengthened, and hence accepted for publication, if for instance, similarly to Refs.[17,18], connections between the entropy production rate and the quantum correlations (over the time evolution and at the steady state) are established.
Minor comments. The wording “axiomatic formalism of the theory of open quantum systems”, is misleading. The adjective “axiomatic” should be removed.
Seminal references about two or more systems dissipating into a common environment should be included.
Round 2
Reviewer 2 Report
The authors have addressed my concerns. In particular they strengthened the work by establishing connections between the entropy production and the quantum correlations. I can support publication of the present version. However, for the sake of readability, the authors could consider to simplify the treatment (and the figures) by restricting to only one quantifier of quantum correlations, instead of considering several of them, like Renyi-2 mutual information and discord and logarithmic negativity.
Author Response
Point 1: The authors have addressed my concerns. In particular they strengthened the work by establishing connections between the entropy production and the quantum correlations. I can support publication of the present version. However, for the sake of readability, the authors could consider to simplify the treatment (and the figures) by restricting to only one quantifier of quantum correlations, instead of considering several of them, like Renyi-2 mutual information and discord and logarithmic negativity.
Response 1: We would like to thank the reviewer for his appreciative comments. According to the recommendation of the reviewer we considered the connection between entropy production rate and only one quantifier of quantum correlations, namely Renyi-2 mutual information, instead of three quantifiers initially analysed in the manuscript. Correspondingly, the content of the manuscript was reduced, including the total number of plots in the figures, which was reduced by approx. 1/3.
This manuscript is a resubmission of an earlier submission. The following is a list of the peer review reports and author responses from that submission.
Round 1
Reviewer 1 Report
I find the manuscript contains no new or interesting physics regarding entropy production in open quantum systems. The theoretical formulation takes shortly from Refs [6, 15, 16] without giving some physical interpretation. In particular, the master equation (4), although it has a standard Lindblad form, looks very strange because the operator B_j in the equation is not specified. It is not clear what kind of diffusion and dissipation phenomena are contained in this master equation. The system (two coupled harmonic oscillators coupled the environment) is already studied in Refs [15,16]. The non-Markovian character in such a formulation of entropy production has also been studied in these references. Thus, a Markovian limit considered in this manuscript seems to be no worth doing. In addition, one also cannot find the connection of the so-called Lyapunov equation (7-9) which was derived from the classical Langevin equation, with master equation (4). In particular, one has to question, where the dissipation rate \lambda and the temperature come from in Eqs (8-9)? The formulas of Sec III are also fully copied from Refs [6,15,16]. If the manuscript contains some things new, they must be the 16 Figures which are corresponding to different parameter setup of the system Hamiltonian in Eq (5) and different initial state taking of the two harmonic oscillators. But no explanation of the physics (if there is any) is given about these figures. In short, the manuscript does not meet the basic requirement for a scientific paper, I do not think it is worth for publication.
Reviewer 2 Report
Authors consider production of entropy for different times of quantum states on the example of Gaussian states; the states include squeezed states, thermal states, coherent states, and other types of Gaussian states. The time evolution of open quantum system states obeyng the GKSL equation and the corresponding evolution of entropy are studied. The results contain the novelty and they are illusttrated by relevant plots. The paper is interesting for readers and can be published in Special issue of Entropy: Dynamics of Quantum Correlations in Open Systems. The only comment is that it is worth shortly mentioning the properties of tomographic entropies of quantum states published, for example, in AIP Conf. Proc., 1334, 217 (2011) and recent paper published in Entropy, 23, 1445 (2021). After such minor revisions, paper can be accepted for publication.